# Effects of Individual Amino Acids on PPARα Transactivation, mTORC1 Activation, ApoA-I Transcription and pro-ApoA-I Secretion

**DOI:** 10.3390/ijms23116071

**Published:** 2022-05-28

**Authors:** Jehad Z. Tayyeb, Herman E. Popeijus, Janna van de Sanden, Willem Zwaan, Ronald P. Mensink, Jogchum Plat

**Affiliations:** 1Department of Nutrition and Movement Sciences, NUTRIM School for Nutrition and Translational Research in Metabolism, Maastricht University, 6229 ET Maastricht, The Netherlands; dr.jehad_tayeb@hotmail.com (J.Z.T.); j.vandesanden@student.maastrichtuniversity.nl (J.v.d.S.); w.zwaan@maastrichtuniversity.nl (W.Z.); r.mensink@maastrichtuniversity.nl (R.P.M.); j.plat@maastrichtuniversity.nl (J.P.); 2Department of Clinical Biochemistry, Faculty of Medicine, University of Jeddah, Jeddah 23218, Saudi Arabia

**Keywords:** ApoA-I, amino acids, PPARα, mTOR

## Abstract

A higher concentration of apolipoprotein A-I (ApoA-I) is associated with increased high density lipoprotein functionality and reverse cholesterol transport (RCT). A promising strategy to prevent cardiovascular diseases is therefore to improve RCT by increasing de novo ApoA-I production. Since experimental animal models have suggested effects of amino acids on hepatic lipoprotein metabolism, we here examined the effects of different amino acids on hepatic ApoA-I production. Human hepatocytes (HepG2) were exposed to six individual amino acids for 48 h. ApoA-I transcription and secreted pro-ApoA-I protein concentrations were analyzed using quantitative polymerase chain reaction (qPCR) and enzyme-linked immunosorbent assays (ELISA), respectively. Additionally, CPT1 and KEAP1 mRNA expression, peroxisome proliferator-activated receptor alpha (PPARα) transactivation, and mechanistic target of rapamycin complex 1 (mTORC1) phosphorylation were determined. Leucine, glutamic acid, and tryptophan increased ApoA-I and CPT1 mRNA expression. Tryptophan also strongly increased PPARα transactivation. Glutamine, proline, and histidine increased pro-ApoA-I protein concentrations but mTORC1 phosphorylation remained unchanged regardless of the amino acid provided. In conclusion, individual amino acids have different effects on ApoA-I mRNA expression and pro-ApoA-I production which can partially be explained by specific effects on PPARα transactivation, while mTORC1 phosphorylation remained unaffected.

## 1. Introduction

Numerous studies have consistently demonstrated a relationship between serum lipid and lipoprotein concentrations with cardiovascular diseases (CVD) [1,2,3]. The best characterized risk factor for CVD is an elevated serum concentration of low-density lipoprotein (LDL) cholesterol. On the other hand, cross-sectionally, high-density lipoprotein (HDL) cholesterol concentrations are inversely associated with CVD [4]. This association most likely relates to the functionality of the HDL particles, since an increased concentration of functional apolipoprotein A-I (ApoA-I), the major component of HDL, results in increased reverse cholesterol transport (RCT) [5,6]. Thus, together with lowering LDL cholesterol, a promising additional strategy to prevent CVD might be to improve RCT by increasing de novo ApoA-I production [7,8].

Besides pharmacological treatment in the case of manifested dyslipidemia, prevention of dyslipidemia at an early stage through adoption of a healthy lifestyle is currently receiving significant attention. This includes, at minimum, maintaining an optimal body weight, implementing less sedentary behavior, and optimizing dietary intake and composition. To increase HDL functionality and ApoA-I concentrations, there has been a strong interest in the effects of increasing the intake of specific dietary fats or fatty acids at the expense of carbohydrates [9,10,11]. Effects of dietary proteins or individual amino acids on serum HDL cholesterol and ApoA-I concentrations have so far not been considered in great detail. However, various animal models have suggested an association between amino acid intake and hepatic lipid metabolism [12,13]. For example, in Sprague-Dawley rats with non-alcoholic fatty liver disease (NAFLD), citrulline supplementation lowered hypertriglyceridemia and reduced hepatic fat accumulation [12]. In ApoE null mice, a diet with extra leucine improved the plasma lipid profile, diminished systemic inflammation, and markedly attenuated atherosclerosis [14]. Finally, in New Zealand white rabbits fed a high cholesterol diet, oral administration of glutamate and aspartate increased serum HDL cholesterol concentrations, maintained plasma ApoA-1 concentrations, and reduced fatty streak formation [15]. The exact effects of individual amino acids on hepatic ApoA-I production or on mechanisms underlying ApoA-I production have not been studied in great detail. Peroxisome proliferator-activated receptor alpha (PPARα) transactivation or bromodomain and extra-terminal (BET) inhibition, for example, are known regulators of ApoA-I gene expression [16,17]. On the other hand, the mechanistic target of rapamycin complex 1 (mTORC1) is known to respond to amino acid deprivation, but its role in regulating ApoA-I transcription and secretion is unclear. However, amino acid depletion was recently shown to reduce de novo ApoA-I production, which was suggested to be due to inhibited mTORC1 activity [18]. Based on these findings, we hypothesized that amino acids can potentially affect hepatic ApoA-I transcription and pro-ApoA-I production. Therefore, we examined the effects of six different amino acids (glutamine, leucine, proline, histidine, glutamic acid, and tryptophan), which were selected based on different physical and chemical characteristics, on ApoA-I transcription and pro-ApoA-I production in HepG2 cells. We also explored the potential roles of PPARα transactivation, BET inhibition, and mTORC1 phosphorylation in the effects of amino acids on ApoA-I production.

## 2. Results

### 2.1. Effects of BET Inhibitor JQ1(+)

JQ1(+) is a BET-inhibitor that upregulates ApoA-I and downregulates Kelch-like ECH-associated protein 1 (KEAP1) gene expression and can, therefore, serve as control for transcriptional processes. As expected, JQ1(+) increased transcription of ApoA-I, and reduced KEAP1 transcription in HepG2 cells, but had no effect on CPT1 gene expression (Figure 1A). JQ1(+) also increased pro-ApoA-I secretion (Figure 1B). This verifies that the HepG2 cells responded well to external stimuli [16]. HepG2 cells generally do not respond well to stimuli with regard to mature ApoA-I production and secretion, hence why we did not include mature ApoA-I data to this paper.

### 2.2. Effects of Different Amino Acids on ApoA-I mRNA Expression and pro-ApoA-I Secretion

Leucine (Leu), glutamic acid (Glu), and tryptophan (Trp) dose-dependently increased ApoA-I mRNA expression (*p* < 0.05). At concentrations of 10 mM Leu and Glu, ApoA-I mRNA expression increased by 32% and 21%, respectively. The most pronounced increase of 47% for Trp was found at a concentration of 5 mM. The other three amino acids, glutamine (Gln), proline (Pro), and histidine (His), did not change ApoA-I mRNA expression (Figure 2A). For the individual amino acids, the patterns for pro-ApoA-I protein concentrations were opposite to those of ApoA-I mRNA expression. Gln, Pro, and His increased pro-ApoA-I protein concentrations dose-dependently (*p* < 0.05), whereas Trp, Leu, and Glu treatment did not affect pro-ApoA-I protein concentrations (Figure 2B). Surprisingly, pro-ApoA-I protein concentrations decreased after Trp treatment (*p* < 0.001).

### 2.3. Effects of Different Amino Acids on BET Inhibition and PPARα Transactivation

Effects of the different amino acids on CPT1 and KEAP1 mRNA expression were first analyzed to explore possible pathways underlying the effects on ApoA-I mRNA expression and protein secretion. mRNA expression of the BET inhibition target gene KEAP1 dose-dependently decreased after Gln treatment (*p* < 0.001), while Trp dose-dependently increased (*p* < 0.001) KEAP1 mRNA expression. Leu, Pro, His, and Glu did not significantly change KEAP1 mRNA expression (Figure 3).

Interestingly, HepG2 cells exposed to Gln, Leu, His, Glu, and Trp consistently showed a significant (*p* < 0.01) increase in mRNA expression of PPARα target gene CPT1. Only Pro did not affect CPT1 mRNA expression (Figure 4A). The association between CPT1 mRNA expression with ApoA-I mRNA expression was calculated, as the ApoA-I promotor contains a PPAR response element as well. An increase in both CPT1 and ApoA-I transcription might, thus, indicate increased PPARα transactivation. A significant and positive association was found after Trp treatment (r = 0.538; *p* < 0.05). Such an association was not observed for the other amino acids that increased ApoA-I mRNA expression. We further examined the involvement of PPARα transactivation by treating transfected cells with Trp or His, the amino acids that most potently increased ApoA-I mRNA expression and pro-ApoA-I secretion, respectively. The experimental medium was supplemented with fetal bovine serum (FBS) but not with non-essential amino acids (NEAA). Trp significantly increased PPARα transactivation dose-dependently (Figure 4B). Treatment with 10 mM of Trp even elevated the transactivation more than seven times compared to control conditions. In line with the absence of an effect on ApoA-I mRNA expression, His did not increase PPARα transactivation. 

### 2.4. Effects of Different Amino Acids on mTORC1

To determine whether differential effects on mTORC1 phosphorylation by the individual amino acids could explain why some amino acids increased pro-ApoA-I secretion and others did not, we performed a western blot using mTOR and p-mTOR (Ser2448) antibodies after treatment with the six amino acids. Figure 5 shows that none of the six amino acids changed the mTOR or p-mTOR levels when compared to the controls at any of the concentrations.

### 2.5. Role of mTORC1 on ApoA-I Transcription and pro-ApoA-I Secretion

To explore the possible role of mTORC1 on pro-ApoA-I production, we evaluated the effects of mTOR activation and inhibition by using MHY1485 and Torin2 as pharmacological mTOR activator and inhibitor, respectively. Neither mTOR inhibition nor mTOR activation significantly affected ApoA-I mRNA expression (Figure 6). Furthermore, pro-ApoA-I protein secretion was only significantly reduced by 50 nM of the mTOR inhibitor but was not affected by other concentrations of the inhibitor or activator (Figure 7). In line with this, western blotting revealed that the inhibitor only influenced p-mTOR expression by reducing the amount of p-mTOR in the cells at concentrations higher than 30 nM, while the activator only induced a small reduction in p-mTOR (Figure 8).

## 3. Discussion

Data from several experimental animal models have suggested effects of proteins or even specific amino acids on hepatic lipid and lipoprotein metabolism. Therefore, we investigated for the first time the effects of different individual amino acids on hepatic ApoA-I transcription and pro-ApoA-I production, and potential underlying mechanisms in HepG2 cells. We showed that the amino acids Leu, Glu, and Trp enhanced ApoA-I mRNA expression, while Gln, Pro, and His associated with elevated pro-ApoA-I protein concentrations in the culture medium. The increase of ApoA-I mRNA was inconsistent with the KEAP1 mRNA concentrations but was mostly consistent with observed CPT1 mRNA concentrations. This implies that the increase in ApoA-I mRNA expression by Trp, Leu and Glu was not mediated by BET inhibition, while the PPARα transactivation pathway seems to play an important role. In further support, CPT1 mRNA expression was significantly positively associated with ApoA-I mRNA expression in response to Trp treatment. This association was not found for the other amino acids, suggesting that Trp could be the most potent PPARα activator, which coincided with the largest elevation in ApoA-I mRNA expression. In follow up experiments we indeed showed that Trp strongly increased PPARα transactivation whereas His (the amino acid that elevated pro-ApoA-I concentrations most pronounced, but not ApoA-I mRNA) did not. These findings imply that the difference in ApoA-I mRNA expression between amino acids can most likely be attributed to different capacities of the amino acids to transactivate PPARα. This is in line with the link between PPARα and ApoA-I expression, which was also found in our previous studies with short chain fatty acids. This is supported by the ability of fatty acids to bind to the ligand binding domain of PPARα and by the presence of a PPAR response element in the promotor region of ApoA-I [19,20]. Next to fatty acids, bilirubin has recently been identified as endogenous ligand for PPARα [21,22]. Similarly, amino acids might be endogenous ligands as well. Interestingly, liver specific PPARα knockout in mice was shown to not affect plasma ApoA-I levels [23], although other research using human liver cells has confirmed that specifically activating PPARα does increase ApoA-I mRNA expression [16,24]. A possible link between amino acids and PPARα transactivation was observed in previous studies using other experimental settings, as well. For example, leucine significantly increased PPARα expression in skeletal muscle myotubes of mice [25]. Also, leucine, glutamine, and proline all reduced IL-8 production in HepG2 cells, probably via NF-κB inhibition [26], a process in which PPARα is also involved [27]. However, in this latter study, PPARα transactivation was not evaluated. Chemical characteristics of the amino acids may have played a role in explaining the selectivity regarding effects on ApoA-I mRNA expression, as the three amino acids that increased ApoA-I expression (Leu, Glu, and Trp) were more hydrophobic than the other amino acids that did not (Gln, Pro, and His). However, it remains to be investigated whether the amino acids have their effects by directly binding to the PPARs, like bilirubin and fatty acids do [20,22], or if they act via modulation of their dimeric partners or through a yet undiscovered route on PPAR transactivation. Additionally, future studies using a PPARα inhibitor are required to confirm whether PPARα transactivation is directly responsible for the observed increase in ApoA-I mRNA after treatment with Trp, Leu, and Glu.

Besides the observation that effects of amino acids on ApoA-I transcription seem mediated via PPARα transactivation, a second finding was that a transcriptional increase of ApoA-I mRNA did not translate into a change in pro-ApoA-I protein concentrations. In contrast, the amino acids Gln, Pro, and His did not affect ApoA-I mRNA expression but increased pro-ApoA-I protein concentrations. It can be speculated that the amino acids have different effects on ApoA-I mRNA stability, which affects the translational capacity of ApoA-I mRNA [28]. Another explanation may relate to mTORC1 activation. Georgila et al. showed that amino acid deprivation inhibited mTORC1 activity and thereby reduced ApoA-I protein levels, but not ApoA-I mRNA expression in HepG2 cells [18]. Moreover, a recent study by Wang et al. showed that a cocktail of leucine, arginine, and lysine increased mTORC1 activation in HL-1 cells [29]. Therefore, we also evaluated the effects of individual amino acids on mTORC1 activity. However, we found that none of the six explored amino acids changed mTORC1 phosphorylation compared to controls and they did not differ in their capacities to phosphorylate mTORC1 when compared to each other. Interestingly, specific mTORC1 activation by using the activator MHY1485 did not further increase ApoA-I mRNA expression or pro-ApoA-I levels. It is important to note that the previously mentioned studies used amino acid deprived cells that were only treated for 1–6 h [18,29]. Our cells were, however, treated for 48 h and were probably not enough deprived of amino acids to reduce the activity of the mTORC1 pathway due to the amino acids already present in the culture medium (Appendix A). The abundance of amino acids might have already maximally activated mTORC1 which potentially explains why the activator and the individual amino acids did not increase mTORC1 phosphorylation or pro-ApoA-I concentrations. In line with previous research [18], inhibition of mTORC1 by using Torin2 did not affect ApoA-I mRNA expression. Interestingly, inhibition also did not significantly reduce pro-ApoA-I secretion, except at 50 nM. Furthermore, we also did not find a clear change in mTOR phosphorylation after inhibition or activation of mTOR. It could be that effects of amino acids on mTOR phosphorylation were no longer present due to the relatively long treatment time (48 h) which might have been too long to observe the effects of mTOR inhibition or activation. Changes in ApoA-I mRNA expression upon mTORC1 activation or inhibition might still be expected since mTORC1 is known to regulate the transcription or activation of proteins involved in lipid metabolism (e.g., SREBP1) and ApoA-I transcription (e.g., PPARα) [30,31,32]. mTORC1 can, for example, inhibit PPARα transactivation which could have been why the amino acids that increased pro-ApoA-I did not show elevated ApoA-I mRNA activation, had they activated mTORC1 [32]. However, more research is required to elucidate how exactly mTORC1 is involved in ApoA-I transcription and pro-ApoA-I synthesis and to understand what happens to amino acid-induced ApoA-I production when mTORC1 is inhibited, while measuring ApoA-I mRNA and pro-ApoA-I levels, and mTOR phosphorylation at shorter time points. Additionally, experiments can be performed using primary hepatocytes to determine the effects of amino acids on the production of mature ApoA-I.

Another mechanism involved in regulation of translation by amino acids that we did not study here involves mammalian general control nonderepressible 2 (mGCN2). mGCN2 is a serine/threonine protein kinase that, in case of amino acid deprivation, phosphorylates eukaryotic initiation factor 2 alpha (eIF2α) which then inhibits initiation of translation [33]. One study showed that livers from mGCN2^+/+^ mice with limited His availability expressed increased eIF2α phosphorylation along with reduced eIF2B activity and reduced protein synthesis [34]. Therefore, perhaps our observed increase in pro-ApoA-I after supplementing HepG2 cells with His is due to a negative effect of His on mGCN2 activity, resulting in lowered eIF2α phosphorylation, increased eIF2B phosphorylation and, consequently, increased translation. Other studies showed that weanling rats fed a diet lacking Leu or Trp expressed lower eIF2B activity and protein synthesis along with increased eIF2α phosphorylation due to increased mGCN2 activity [35,36]. It can be expected that supplementation with Leu or Trp would, in turn, increase protein synthesis. However, our findings argue against that as we found that Leu and Trp did not increase pro-ApoA-I levels despite increasing ApoA-I mRNA expression. Regardless, findings from these studies suggest that an amino acid imbalance influences translation through mGCN2 [33]. Therefore, we propose that future research should also focus on the effects of individual amino acids on the mGCN2 pathway and the potential link with ApoA-I production to understand why pro-ApoA-I increased for some amino acids but not for others. 

In conclusion, individual amino acids have different effects on ApoA-I gene expression and pro-ApoA-I protein secretion. Leucine, glutamic acid, and tryptophan increased ApoA-I and CPT1 mRNA expression in HepG2 cells. Tryptophan also strongly induced PPARα transactivation while histidine did not. Further research is warranted to acquire better understanding of the differential effects of amino acids on PPARα activity. The increase in ApoA-I mRNA expression did, however, not translate into elevated pro-ApoA-I concentrations. Interestingly, glutamine, proline, and histidine increased pro-ApoA-I concentrations without elevating ApoA-I mRNA expression. We showed that this discrepancy between the two categories of amino acids cannot be explained by differences in the capacity of activating mTORC1 by the amino acids. A start to understand the mechanisms behind the effects of amino acids on ApoA-I mRNA and pro-ApoA-I expression has been made. Of course, more research is required to elucidate the fundamental processes involved and should also focus on effects on mature ApoA-I production and secretion in different cell lines at different time points.

## 4. Materials and Methods

### 4.1. Materials

Human hepatocellular liver carcinoma cells (HepG2) were kindly provided by Sten Braesch-Andersen (Mabtech, Nacka Strand, Sweden). Cell culture flasks and plates were obtained from Corning (Cambridge, MA, USA). Minimum Essential Medium (MEM), sodium pyruvate, non-essential amino acids (NEAA), penicillin, and streptomycin were all obtained from Thermo Fisher Scientific (Bleiswijk, Netherlands). Fetal bovine serum (FBS) was purchased from PAA (Toronto, Canada). Glutamine (Gln), leucine (Leu), proline (Pro), histidine (His), glutamic acid (Glu), tryptophan (Trp), dimethyl sulfoxide (DMSO), Tri-reagent, Torin2, and MHY1485 were all bought from Sigma (Uithoorn, Netherlands). The BET inhibitor JQ1(+) was obtained from Bio-Techne—R&D (Minneapolis, MN, USA). The chemical characteristics of the six amino acids used to treat the cells are presented in Appendix A.

### 4.2. Cell Culture and Amino Acids Treatment

HepG2 cells were cultured at 37 °C in a humidified atmosphere of 5% carbon dioxide (CO_2_) in MEM containing 10% heat inactivated FBS, 1% sodium pyruvate, 1% NEAA, and 1% of penicillin-streptomycin mixture. For all experiments, cells were seeded in 24-well plates at a density of 200,000 cells per well. Cell viability was inspected daily by microscope and when cells reached a density of 80–90%, they were incubated for 48 h in experimental medium (MEM without FBS and without NEAA) enriched with a concentration range of 0–10 mM amino acids (Gln, Leu, Pro, His, Glu, or Trp), 3 µM JQ1(+), 0–100 µM MHY1485, or 0–1000 nM Torin2. Effects were expressed relative to those of the non-enriched background medium. BET inhibitor JQ1(+) was included as positive control in each experiment to ensure that cells were responsive and produced sufficient amounts of ApoA-I [16]. JQ1(+), mTOR inhibitor Torin2, and mTOR activator MHY1485 were dissolved in DMSO and their effects were expressed relative to their carrier control. The final DMSO concentration was kept the same in all conditions at 0.2%. Culture medium was collected for analysis of pro-ApoA-I concentrations. Cells used for analysis of mRNA expression were lysed in Tri-reagent as described before [16], or according to the instructions of the Rneasy mini kit (Qiagen, Hilden, Germany). Both culture medium and lysed cells were snap frozen in liquid nitrogen and stored at −80 °C until further analysis.

### 4.3. Quantification of Gene mRNA Transcription

For cDNA synthesis, 350 ng of total RNA was reverse-transcribed using MMLV reverse transcriptase, dNTP’s, random hexamers, DTT, and 5xFS buffer supplemented with RNAse inhibitor (Thermo Fisher Scientific, Bleiswijk, Netherlands). The resulting cDNA was used for real time quantitative PCR using TaqMan Gene Expression Assays using Cyclophilin A (Hs99999904) as a housekeeping control. To quantify mRNA expression of ApoA-I, PPARα target CPT1 and the BET inhibitor target KEAP1, the TaqMan Gene Expression Assays Hs00163641, Hs00912671, and Hs00202227 were used. Values are presented as relative gene expressions based on the Ct values, normalized for the internal control Cyclophilin A, and compared to the control conditions.

### 4.4. Quantification of pro-ApoA-I Protein Levels in the Culture Medium

pro-ApoA-I protein concentrations in culture medium of HepG2 cells were measured by an enzyme-linked immunoassay (ELISA). The pro-ApoA-I ELISA was performed as described [8]. For the pro-ApoA-I ELISA measurements, the absorbances were determined at 450 nm using a multi-scan microplate reader (Thermo Fisher Scientific, Bleiswijk, Netherlands). Values are presented as relative fold changes compared to the control conditions.

### 4.5. Transfection and Measurement of PPARα Transactivation

To determine the involvement of PPARα transactivation, cells were co-transfected with pcDNA3.1 and a PPAR response element cloned in front of a reporter gene luciferase (pGL3_PPRE) [37], using X-treme gene 9 or Fugene 6 transfection reagents according to the manufacturer’s instructions (Roche Diagnostics, Basel, Switzerland) 24 h after plating the cells. A green fluorescent protein vector was used to control the efficiency of the transfection. Another 24 h later, culture medium was changed with medium without NEAA but supplemented with 10% FBS with addition of 0–10 mM of Trp or His for 48 h. After collecting the medium, cells were harvested using a luciferase lysis buffer (Promega, Madison, WI, USA), snap frozen and stored at −80 °C. The cells were later used to measure luciferase activity, reflecting PPARα transactivation, using a luciferase assay substrate (Promega, Madison, WI, USA) and the GloMax 96 Microplate luminometer (Promega, Madison, WI, USA).

### 4.6. Western Blot for mTORC1 Activity

mTORC1 activity is reflected by mTORC1 phosphorylation which was determined by western blot as previously described [38]. In short, the treated cells were first lysed with radioimmunoprecipitation assay (RIPA) buffer consisting of 20 mM Tris (pH 7.4), 150 mM NaCl, 1% Nonidet, 1 mM DTT, 1 mM Na vanadate, 1 mM PMSF, 10 µg/mL leupeptin, 1% aprotinin (Sigma-Aldrich). Proteins were separated by electrophoresis using precast Tris acetate gels and XT-tricine buffer (Bio-Rad, Hercules, CA, USA), run at 150 V. The proteins were transferred to nitrocellulose membranes (Bio-Rad) by electroblotting at 100 V for 1 h using the Trans-Blot Turbo system (Bio-Rad). Equal protein loading was checked with PonceauS staining after which the membranes were blocked and overnight probed with the primary antibodies phospho-mTOR-Ser2448 and mTOR from Cell Signaling (Beverly, MA) at 4 °C. Following the secondary antibody and successive washing, protein quantification was performed by scanning using the Odyssey Infrared Imaging System (LI-COR Biotechnology, Lincoln, NE).

### 4.7. Statistical Analysis

All independent dose-response experiments with amino acids were performed in triplicate or quadruplicate and each experiment was repeated at least twice, while the experiments regarding mTOR were performed three times. Regression analyses were used to examine dose-response relationships between the concentrations of added amino acids or mTOR inhibitor or activator and the respective parameters. For the correlations between CPT1 mRNA expression and ApoA-I mRNA expression, Spearman correlations were calculated. The regression coefficients and Spearman correlation coefficients were considered to be statistically significant when different from zero at *p* < 0.05. The effects of individual doses were statistically evaluated versus control conditions by Mann–Whitney U tests. Again, a *p*-value < 0.05 was considered to be statistically significant. All results are presented as means, with error bars indicating standard deviations (SD) or standard error of the mean (SEM). All statistical analyses were performed using SPSS v.25 (IBM Corp., Armonk, NY, USA).

## Figures and Tables

**Figure 1 ijms-23-06071-f001:**
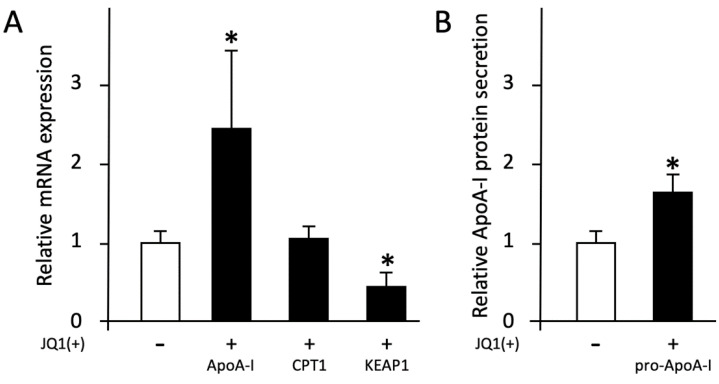
The cells are responsive to treatment. HepG2 cells were treated with BET inhibitor JQ1(+) (3µM) for 48 h as positive control indicated with the “+” or no JQ(+) indicated with the “-“. (**A**) Relative ApoA-I, CPT1, and KEAP1 mRNA expressions measured by qPCR. (**B**) Relative pro-ApoA-I protein secretions measured by ELISA. Results are presented as means with error bars indicating standard deviations. Data were normalized against the expression observed in the control condition, which was arbitrarily set at 1. This experiment was performed twice with each condition in quadruplicate. Differences between JQ1(+) and controls were tested with Mann-Whitney U tests. Significant difference (*p* < 0.05) are marked with an asterisk.

**Figure 2 ijms-23-06071-f002:**
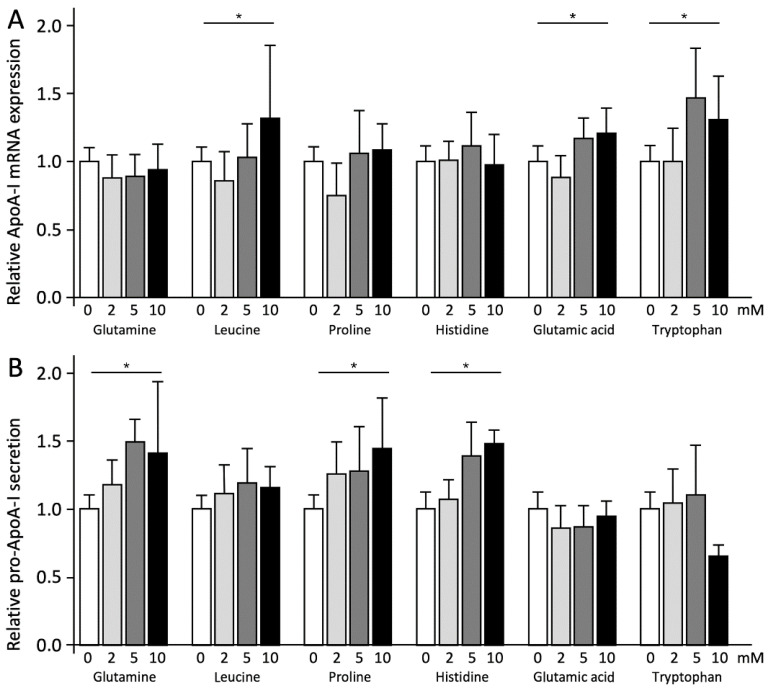
Amino acids have different effects on ApoA-I mRNA expression and pro-ApoA-I secretion. HepG2 cells were treated with 0–10 mM of six different amino acids for 48 h. (**A**) Relative ApoA-I mRNA expressions measured by qPCR. (**B**) Relative pro-ApoA-I protein secretions measured by ELISA. Results are presented as means with error bars indicating standard deviations. Data were normalized against the expression observed in the control condition, which was arbitrarily set at 1. This experiment was performed twice with each condition in quadruplicate. Differences between amino acid treatment and controls were tested with linear regression analyses and were considered significant when the regression coefficients differed from zero (*p* < 0.05). Significant dose-response relations (*p* < 0.05) are indicated by an asterisk above a horizontal line.

**Figure 3 ijms-23-06071-f003:**
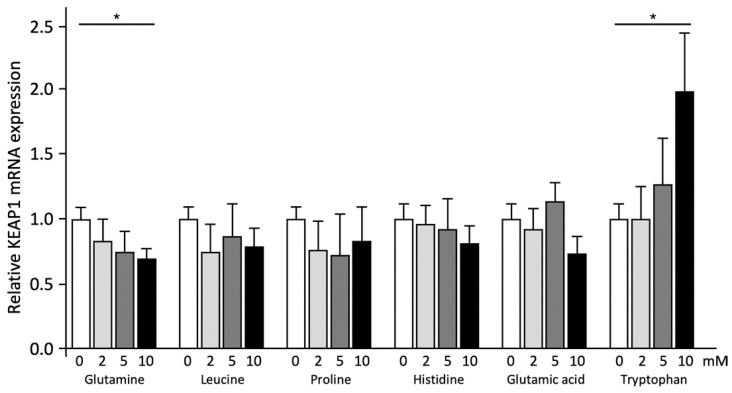
Amino acids have different effects on KEAP1 mRNA expression. HepG2 cells were treated with 0–10 mM of six different amino acids for 48 h and KEAP1 mRNA expression was measured by qPCR. Results are presented as means with error bars indicating standard deviations. Data were normalized against the expression observed in the control condition, which was arbitrarily set at 1. This experiment was performed twice with each condition in quadruplicate. Differences between amino acid treatment and controls were tested with linear regression analyses and were considered significant when the regression coefficients differed from zero (*p* < 0.05). Significant dose-response relations (*p* < 0.05) are indicated by an asterisk above a horizontal line.

**Figure 4 ijms-23-06071-f004:**
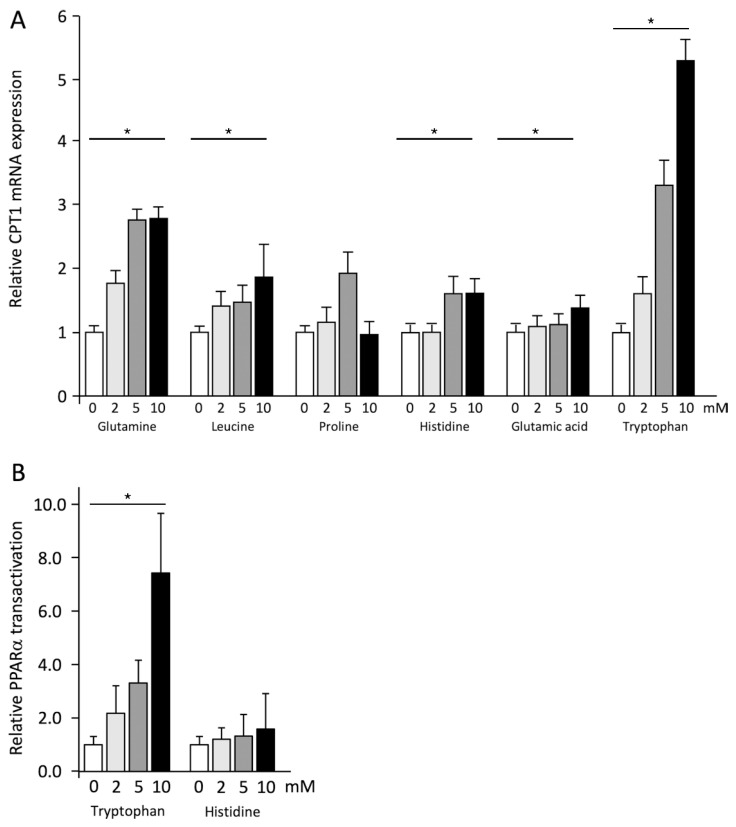
Amino acids have different effects on CPT1 mRNA expression and PPARα transactivation. (**A**) CPT1 mRNA expression in HepG2 cells treated for 48 h with 0–10 mM of six amino acids, measured by qPCR. (**B**) PPARα transactivation in HepG2 cells that were co-transfected with a pcDNA3.1 and pGL3_PPRE vector; 24 h after transfection, cells were treated with 0–10 mM of tryptophan or histidine in medium supplemented with 10% FBS for 48 h, after which cells were lysed. Lysates were used in a luciferase assay to determine PPARα transactivation. Results are presented as means with error bars indicating standard deviations. Data were normalized against the expression observed in the control condition, which was arbitrarily set at 1. The experiment in panel A was performed twice with each condition in quadruplicate and the experiment in panel B was performed three times with each condition in triplicate. Differences between amino acid treatment and controls were tested with linear regression analyses and were considered significant when the regression coefficients differed from zero (*p* < 0.05). Significant dose-response relations (*p* < 0.05) are indicated by an asterisk above a horizontal line.

**Figure 5 ijms-23-06071-f005:**
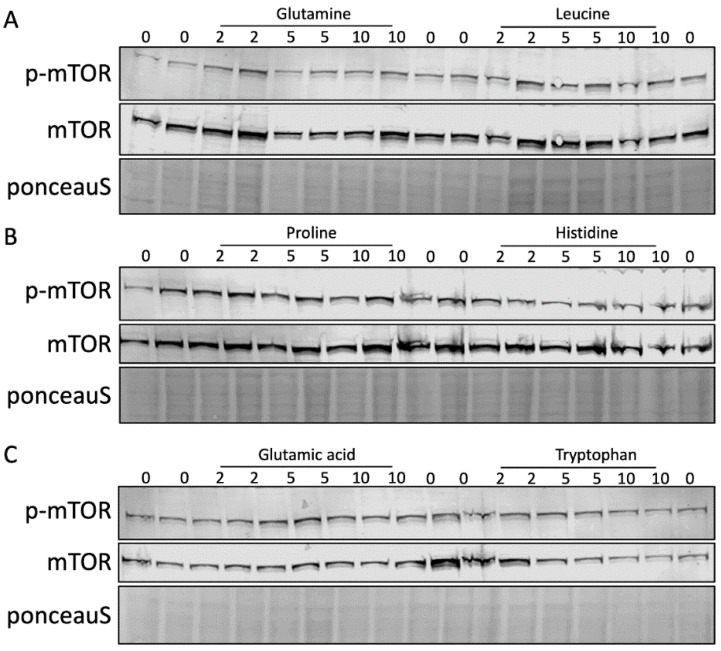
The amino acids do not increase mTOR activity. HepG2 cells were treated with 0–10 mM of six different amino acids for 48 h. mTOR expression and activation was determined by western blotting using mTOR and p-mTOR antibodies with p-mTOR reflecting activated mTOR. Effects of (**A**) glutamine, leucine, (**B**) proline, histidine, (**C**) glutamic acid, and tryptophan. This experiment was performed twice with each condition in duplo.

**Figure 6 ijms-23-06071-f006:**
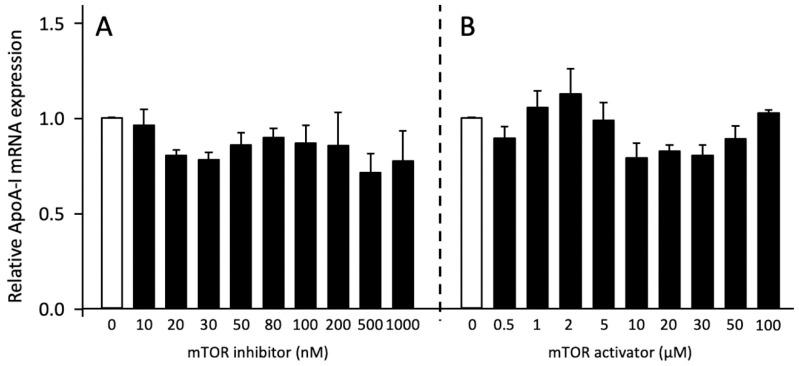
Effects of mTOR inhibition and activation on ApoA-I mRNA expression. HepG2 cells were treated with 0–1000 nM of mTOR inhibitor Torin2 or 0–50 µM of mTOR activator MHY1485 for 48 h. ApoA-I mRNA expression was determined by qPCR. (**A**) Effect of mTOR inhibition. (**B**) Effect of mTOR activation. Results are presented as means, with error bars indicating standard errors. Data were normalized against the expression observed in the control condition, which was arbitrarily set at 1. This experiment was performed three times with each condition measured in duplo. Dose-dependent effects were tested by linear regression analysis and effects of single doses were compared to controls using Mann-Whitney U tests in case of insignificant dose-response effects. Changes were considered significant for *p* < 0.05.

**Figure 7 ijms-23-06071-f007:**
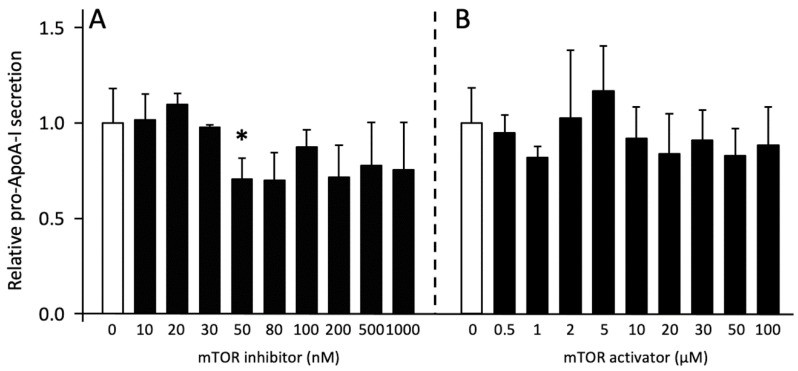
Effects of mTOR inhibition and activation on pro-ApoA-I secretion. HepG2 cells were treated with 0–1000 nM of mTOR inhibitor Torin2 or 0–100 µM of mTOR activator MHY1485 for 48 h. pro-ApoA-I secretion was determined by ELISA. (**A**) Effect of mTOR inhibition. (**B**) Effect of mTOR activation. Results are presented as means, with error bars indicating standard errors. Data were normalized against the expression observed in the control condition, which was arbitrarily set at 1. This experiment was performed three times and with each condition measured in duplo. Dose-dependent effects were tested by linear regression analysis and effects of single doses were compared to controls using Mann-Whitney U tests in case of insignificant dose-response effects. Changes were considered significant for *p* < 0.05. Significant dose-response relations (*p* < 0.05) are indicated by an asterisk above a horizontal line and significant individual doses are indicated by single asterisks.

**Figure 8 ijms-23-06071-f008:**
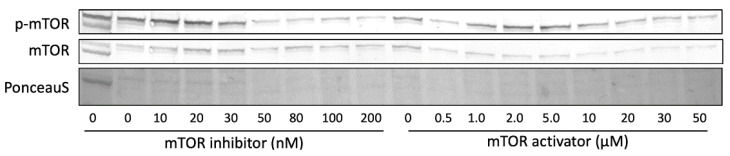
Effects of mTOR inhibition and activation on mTOR phosphorylation. HepG2 cells were treated with 0–200 nM of mTOR inhibitor Torin2 or 0–50 µM of mTOR activator MHY1485 for 48 h. mTOR expression and phosphorylation was determined by western blotting using mTOR and p-mTOR antibodies with p-mTOR reflecting activated mTOR. This experiment was performed three times with each condition in duplo.

## Data Availability

The data presented in this study can be requested from the corresponding author.

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
