# Peer review of "Effects of Individual Amino Acids on PPARα Transactivation, mTORC1 Activation, ApoA-I Transcription and pro-ApoA-I Secretion"

_ijms, 2022, doi:10.3390/ijms23116071_

Round 1
Reviewer 1 Report
The authors of the paper aimed to study the effect of certain amino acids upon ApoA-1 mRNA expression and secretion. They suggest effects found on ApoA1 are probably via PPARa for some amino acids to increase pro-ApoA-1 secretion (e.g Tryptophan) and for other amino acids the effect could be mediated by differences in the capacity of activating mTORC1. Despite no statistical difference upon amino acid treatment were found in mTOR and p-mTOR levels in a single experiment, the authors show a significant effect of the mTOR pathway in apoA1 production. Besides the analysis of possible mechanisms to regulate Apoa1 production, the authors find no difference in mature ApoA-1 levels. The authors present a good discussion stating all limitations of the experimental results and design. Although, it’s a good start for understanding the effects amino acid could have on ApoA-1 levels and circulating lipid and discovering the molecular pathways to those effects, more validation of the results presented is needed.
Major issues
Lack of evidence of the mechanism in vivo, the authors confirm that “Interestingly, liver-specific PPARα knockout in mice was shown to not affect plasma ApoA-I levels, although other research using human liver cells has confirmed that specifically activating PPARα does increase ApoA-I mRNA expression”. The cell system the authors use comes from immortalized cell line and may have different gene expression profiles and transcription mechanisms. Using a primary cell culture system to confirm their findings or supplementing aa (at least tryptophan and histidine) to mouse diet would be necessary to have a confirmation of the mechanistic insight the authors found.
Significance of results? Given that no differences in mature apoa1 were observed there are doubts on the relevance of the effect the amino acids have on mature ApoA1 production overall. Since the authors find a slight reduction of mature ApoaA-1 I suggest evaluating mature ApoA-1 secretion in complete media at earlier time points in case of misfolding or degradation is taking place. Alternatively, it will be good to try detecting mature ApoA1 levels in cell lysates in case apoa1 is not secreted or bound to the membrane. Also, as shown by the authors in figure 1B, Jq1 control does not seem to increase much mature apa1 secretion either. So, the lack of a significant observation of ApoA1 may come from the cell system the authors are using, given that they tested all other factors that could be included in the media.
As authors stated in their limitations, 2 independent experiments are ideal to confirm the potential role mTOR has in ApoA-I production. This limitation can be addressed since the authors already have the cell system and assay validated. The results using mTOR inhibition and activator suggest mTOR is engaged in pro-APA1 secretion rather than expression. From the graph it looks like total mTOR and some p-mTOR is decreased at least on Histidine and these may confirm both that PPARa transactivation is the pathway responsible for tryptophan effects on ApoA1 and mTOR mechanisms are engaged in increasing ApoA1 secretion as authors show for histidine. Also, evaluation of phosphorylation at earlier timepoints may better reflect the effects upon expression or release of ApoA1 since at 48 h the output of expression and secretion has already taken place and the mTOR effect could happen way before that. Confirming the role of the mTOR pathway in histidine and tryptophan effect is necessary and relevant given the results the authors highlight for mTOR in ApoA1 production, this ought to be done in FBS supplemented media.
Minor issues
Please add n and if any number of independent experiments were performed in the figure legends.
From the graphs, it looks like only the highest concentration of amino acids has a significant effect. If other concentrations are significant change the representation add more lines or another strategy to specify which data are statistically different.
Line 76. a brief introduction/description for CPT1 and KEAP and why the authors used them will be better appreciated.
Author Response
Thank you for your critical review we have adapted the manuscript accordingly.

Reviewer 2 Report
This manuscript investigated whether the amino acids (glutamine, leucine, proline, histidine, glutamate and tryptophan) could stimulate APOA-I production that may help to increase HDL and prevent cardiovascular diseases. There are some comments for the authors.
- HepG2 is a liver carcinoma cell line and cancer cells usually have distinct metabolic behavior. The results in this study cannot resemble the effects of these amino acids in normal hepatic cells. Is there any normal hepatic cells can be applied in this study to prove that normal liver cells present the same results as HepG2 did.
- Luciferase assay to determine PPARα transactivation is not a direct evidence to prove that tryptophan and histidine regulate APOA-1 production via PPARa. A PPARa inhibitor group should be added to observe whether the inhibitor could block APOA-1 production.
- The time of treatment for 48 hours may be too long to observe mTOR phosphorylation. Did the authors observe the phosphorylation at different time points?
- Figure 5 and figure 9, how many repeats of the western blotting were performed? How to confirm that the results from one-time experiment could be trusted? The blots should be quantified.
- How many repeats were performed in figure 6? There is no standard deviation on the bars.
Author Response

(The authors gave the same response as above.)

Round 2
Reviewer 1 Report
The authors have edited the manuscript according to comments. I agree with the edits and particularly like the approach of removing the mature ApoA1 results. the line numbering in the formatted version that I received does not correspond to the line references in the responses and there are some lines missing and could not find the statement the authors included in the main text regarding future directions related to mature ApoA1 secretion (minor technical issue, probably journal formatting issue).
It is understandable that the goal was to see if mTOR has the potential to regulate ApoA1 production. However, total and phosphorylated mTOR is a key result to understanding the mechanism by which the aminoacids regulate pro-ApoA1 secretion. so, I still think repeating these experiments would confirm the inhibitory effect upon activation of mTOR, and for the aminoacids would confirm at least histidine's effect upon ApoA1 expression and PPARa transactivation. I would ask editors to allow more time for the performance of these experiments.
Author Response
Thank you for the comments we repeated some experiments as requested and adapted the manuscript accordingly.

Reviewer 2 Report
I have no further comment.
Author Response
Thank you for accepting our paper. You clearly helpt us improving the manuscript.